# Device-Independent Certification of Maximal Randomness from Pure Entangled Two-Qutrit States Using Non-Projective Measurements

**DOI:** 10.3390/e24030350

**Published:** 2022-02-28

**Authors:** Jakub J. Borkała, Chellasamy Jebarathinam, Shubhayan Sarkar, Remigiusz Augusiak

**Affiliations:** Center for Theoretical Physics, Polish Academy of Sciences, Aleja Lotników 32/46, 02-668 Warsaw, Poland; jebarathinam@cft.edu.pl (C.J.); sarkar@cft.edu.pl (S.S.); augusiak@gmail.com (R.A.)

**Keywords:** randomness certification, self-testing, extremal POVM, Weyl–Heisenberg basis

## Abstract

While it has recently been demonstrated how to certify the maximal amount of randomness from any pure two-qubit entangled state in a device-independent way, the problem of optimal randomness certification from entangled states of higher local dimension remains open. Here we introduce a method for device-independent certification of the maximal possible amount of $2\log_{2}3$ random bits using pure bipartite entangled two-qutrit states and extremal nine-outcome general non-projective measurements. To this aim, we exploit a device-independent method for certification of the full Weyl–Heisenberg basis in three-dimensional Hilbert spaces together with a one-sided device-independent method for certification of two-qutrit partially entangled states.

## 1. Introduction

The intrinsic randomness of quantum theory manifested in the outcomes of quantum measurement is one of the most intriguing features of quantum mechanics [1]. Even more remarkable is the fact that quantum technologies allow us to generate certifiable randomness with an unprecedented level of security [2]. Protocols designed for randomness certification ensure both the generation of completely random bits and their privacy, which for instance introduces new possibilities in designing protocols for tasks such as quantum cryptography and quantum key distribution [3].

Since the pioneering works on randomness certification [4] (see also Ref. [2]), significant progress has been made both in theoretical and experimental aspects [5,6,7,8]. It was shown, for instance, in Ref. [9], that maximal violation of the Salavrakos–Augusiak–Tura–Wittek–Acín–Pironio (SATWAP) Bell inequality [10] enables self-testing the maximally entangled state of two-qudits of arbitrary local dimension, which in turn allows certifying $\log_{2}d$ bits of randomness by using projective measurements. On the other hand, we know that non-projective measurements, also known as positive-operator valued measures (POVM), can be used to generate more randomness in a given dimension than projective ones. The intuitive reason behind this is the existence of extremal $d^{2}$-outcome non-projective measurements in *d*-dimensional Hilbert spaces, which consequently might give rise to $2\log_{2}d$ random bits [11]. In fact, a method for certification of two bits of local randomness in dimension two by exploiting such non-projective measurements was introduced in Ref. [12].

Similar research was conducted in Ref. [13], where the authors exploited Gisin’s elegant Bell inequality [14] instead of the Clauser–Horne–Shimony–Holt (CHSH) inequalities used in Ref. [12]. Later, in Ref. [15] it was shown how to certify the maximal amount of local randomness independently of the degree of entanglement of two-qubit states.

While significant progress has been made in understanding the possibility of device-independent randomness certification from entangled states of the lowest possible dimension, higher-dimensional scenarios remain mostly unexplored; see nevertheless the recent work [16] presenting an approach in which by using symmetric informationally complete POVMs one can obtain more than $\log_{2}d$ bits of local randomness from the maximally entangled two-qudit state of the local dimension up to d=7.

However, it remains an open and highly nontrivial problem whether it is possible to device-independently certify the maximal amount of $2\log_{2}d$ bits of randomness by performing measurements on quantum systems of dimension *d* for any finite *d*. Another interesting direction to explore is whether the maximal amount of randomness can be certified independently of the degree of entanglement of states used in the protocol. In this work considering d=3, we provide a positive answer to the first problem and a partial solution to the second one; that is, we show how to certify $2\log_{2}3$ bits of randomness by performing local non-projective measurements on a well-defined subset of pure bipartite entangled states in a fully device-independent way.

In our work, we use the family of Bell inequalities proposed in Ref. [17] that allows for self-testing the two-qutrit maximally entangled state as well as three mutually unbiased bases (MUBs) per party. We extend the self-testing proof of [17] to certify, up to the transposition equivalence, all of the Weyl–Heisenberg (W-H) operators acting on three-dimensional Hilbert spaces. Our approach for W-H basis certification is inspired by Ref. [15] and is based on simultaneous maximal violation of the Bell inequality from Ref. [17] by two appropriately selected sets of measurements. In this way, we can self-test a complete set of four MUBs, which allows us to construct eight W-H operators in dimension three. As a consequence of certifying the complete W-H basis, we can characterize any measurement acting on three-dimensional Hilbert space in terms of basis elements [18]. Let us note that our self-testing statements for measurements are always up to the standard equivalences [19], but also up to the transposition equivalence.

The structure of our article is the following. In the preliminaries, we first present the scenario used in our work. Next, we review the Bell inequality from [17], with slight modifications that are necessary for presenting our results. Afterwards, we also review the steering inequality introduced in Ref. [20], which, together with the Bell inequality [17] enables certification of any pure bipartite entangled state of local dimension three. In the second part, where we present our results, we provide a method for device-independent certification of the full W-H basis. Then, we present the main result of our work, which is proof for device-independent certification of a maximal amount of local randomness from pure entangled states in dimension three. Finally, we recall the construction of extremal qutrit POVM of Ref. [20] for a significant subset of partially entangled states that can be used for optimal randomness certification.

## 2. Preliminaries

### 2.1. Scenario

Since we are concerned with the device-independent certification of randomness, we consider an adversarial Bell scenario, consisting of two parties, Alice and Bob, and an adversary, Eve. Alice and Bob cannot trust their devices in this scheme because Eve, a malicious eavesdropper, could have full control of all of their resources. An example of Eve’s strategy might be to use extra dimensions of the Hilbert space hidden in the devices to learn about the results of Alice’s and Bob’s measurements. Eve may also try to entangle with the subsystems of our protagonists Alice and Bob and create correlations, which will allow her to obtain some information on the outputs of the experiment. Nevertheless, the strength of the randomness certification techniques lies in the possibility to prove that, despite any attacks, Eve cannot learn anything about the results of Alice’s and Bob’s measurements. Security of the protocol is demonstrated if they both observe strong correlations in their measurement statistics, i.e., correlations that exhibit the maximal quantum violation of a given Bell inequality.

We construct the following scenario to certify randomness from pure bipartite entangled states of local dimension three in a device-independent way. Alice and Bob perform local measurements on their quantum subsystems, labelled by *A* and *B*, which they receive from the preparation device P operated by Charlie and consisting of two inputs p=1,2. Preparation $P_{1}$ corresponds to preparing a state $\rho_{AB}^{1}$ and $P_{2}$ a state $\rho_{AB}^{2}$. Both preparations can be purified as $|\Psi_{ABE}^{1}\rangle$ and $|\Psi_{ABE}^{2}\rangle$, respectively. Charlie can freely choose the input of the preparation device.

Alice’s device has nine inputs labelled by j=0,⋯,8 and Bob’s device has four inputs labelled by k=0,1,2,3. The first eight measurements of Alice and all measurements of Bob result in three outputs, labelled by *a* for Alice and *b* for Bob such that a,b=0,1,2. The ninth measurement on Alice’s side corresponding to j=8 is a nine-outcome measurement. We employ this additional measurement to certify randomness from its outcomes. A schematic representation of our scenario is presented in Figure 1. It is necessary here to assume that the measurements are independent of the input of the preparation device.

Alice, Bob, and Charlie now collect statistics for each input and the corresponding outputs, which allows them to reconstruct the probability distribution $\vec{p}=\left\{p\left(a,b|j,k,p\right)\right\}$, where p(a,b|j,k,p) is the probability that outcomes *a* and *b* are obtained when performing measurements *j* and *k* on the subsystems *A* and *B* given the preparation *p*. Using this scenario, one can first certify the full Weyl–Heisenberg basis (Section 3.1), then any entangled state of local dimension d=3, and, finally, the optimal amount of randomness from entangled states of local dimension d=3 (Section 3.2).

Let us now discuss potential strategies for Eve that need to be taken into account to ensure that the generated randomness is not accessible by her. For example, suppose Alice wants to generate randomness from the outcomes of one of her measurements. Then the aim for Eve is to guess Alice’s outputs with the highest possible probability. To do so, Eve can prepare Alice’s and Bob’s systems in any way compatible with the given statistics $\vec{p}$ by using quantum resources while remaining undetected. Therefore, we can characterize Eve’s strategy *S* applied for the attack using four major points:1.Eve, for instance, may know the input of the preparation device *p* and the inputs of Alice and Bob x,y, but she cannot change them.2.Eve might possess some subsystem *E* correlated with both parties. Consequently, the state shared among Alice and Bob is defined by $\rho_{AB}={\mathrm{Tr}}_{E}\left(\rho_{ABE}\right)$, where $\rho_{ABE}\in H_{A}⊗H_{B}⊗H_{E}$ denotes the state shared among Alice, Bob, and Eve such that the local Hilbert spaces can be of any arbitrary dimension.3.Eve might have control over Alice’s and Bob’s measurement devices, that is, POVM $F_{j}=\left\{F_{a|j}\right\}$ on $H_{A}$ and POVM $G_{k}=\left\{G_{b|k}\right\}$ on $H_{B}$, respectively.4.Eve’s device is characterized by POVM $Z=\{Z_{a}\}$ on $H_{E}$. The probability of obtaining outcome *a* from a measurement performed by Eve on her share of the joint state $\rho_{ABE}$ is the best guess of Alice’s outcome *a*.

Since there is no restriction on Eve’s subsystem, we can safely assume here that $\rho_{ABE}$ is pure and write $\rho_{ABE}=|\Psi_{ABE}\rangle \;\;\langle \Psi_{ABE}|$. Eve’s influence remains undetected if it cannot be observed in the statistics $\vec{p}$ obtained by Alice and Bob, i.e.,
(1)$$p\left(ab|jk\right)=\langle \Psi_{ABE}|F_{a|j}⊗G_{b|k}⊗{𝟙}_{E}|\Psi_{ABE}\rangle .$$

Now let us define the local guessing probability, that is, the probability that Eve’s guess agrees with Alice’s output [12]:(2)$$G\left(j,\vec{p}\right)=\underset{S\in S_{\vec{p}}}{\sup}\sum_{a}\langle \Psi_{ABE}|\left(F_{a|j}⊗{𝟙}_{B}⊗Z_{a}\right)|\Psi_{ABE}\rangle ,$$
where the supremum is taken over all strategies $S_{\vec{p}}$, consisting of the shared state $\rho_{ABE}$, Bob’s measurements $\left\{G_{b}^{k}\right\}$, and Eve’s measurements $\left\{Z_{a}\right\}$ that reproduce the statistics $\vec{p}$. The amount of random bits obtained from Bob’s measurements is quantified with the min-entropy of the guessing probability $H_{\min}=-\log_{2}G\left(j,\vec{p}\right)$.

Now let us say more about the additional ninth measurement on Alice’s side. We can assume it to be a nine-outcome POVM $\left\{R_{a}\right\}$, which, applied on Alice’s part of an entangled qutrit state, gives completely random results, i.e., $\mathrm{Tr}[R_{a}\rho_{A}]=1/9$, ∀a. Apart from the above conditions, we require that POVM $\left\{R_{a}\right\}$ should reproduce the statistics given by Equation (Equation 1), that is, any of Eve’s attempt of learning of Alice’s outputs remains undetected. We present an example of a measurement construction meeting the above properties in the Section 3.3. Our goal is to prove that Eve’s guessing probability related to Alice’s *j*-th input and consistent with the statistics $\vec{p}$, does not allow Eve to learn anything about Alice’s outputs, i.e., $G(j,\vec{p})=1/9$. Such a situation will provide us with $2\log_{2}3$ bits of private randomness. To sum up, we certify randomness based on the correlations, which minimize Eve’s guessing probability, and these correlations are obtained by measuring POVM $\left\{R_{a}\right\}$ on Alice’s subsystem. For that purpose, we employ an arbitrary entangled state of local dimension 3 certified with the extended Bell scenario and the W-H operators on Bob’s side certified with the use of a Bell test.

### 2.2. Non-Local Scenario and Bell Inequality

In Ref. [17], the authors presented a modification of the Buhrmann–Massar Bell inequality [21] to show self-testing of the maximally entangled state of two-qutrits and three mutually unbiased bases on each site. In our work, we apply this certification scenario to self-test the complete set of W-H operators in dimension three. Later this result is used for the certification of randomness from the measurement $F_{8}=\left\{R_{a}\right\}$. The main idea behind self-testing of the full W-H basis is taken from Refs. [12,15] and consists in using the Bell inequality from Ref. [17] twice. Below we introduce crucial elements needed to present self-testing results.

Throughout this work, we use the correlation picture or, equivalently, the observable picture to describe all correlations observed between Alice and Bob, i.e.,
(3)$$\langle A_{l|j}B_{m|k}\rangle =\sum_{a,b=0}^{2}\omega^{al+bm}p\left(ab|jk\right),$$
where ω=exp(2π𝕚/3) and l,m=0,1,2. The above formula is a two-dimensional Fourier transform of the conditional probabilities p(ab|jk). Operators $A_{l|j},B_{m|k}$ provide us with an alternative description of the measurements $\left\{F_{a|j}\right\}$ and $\left\{G_{b|k}\right\}$ [17], and are defined by Fourier transform in the following way:(4)$$A_{l|j}=\sum_{a=0}^{2}\omega^{al}F_{a|j},\;B_{m|k}=\sum_{b=0}^{2}\omega^{bm}G_{b|k}.$$

Let us note here that we make no assumptions about Alice’s and Bob’s measurements or the shared state; in fact, we consider a fully general situation of $\rho_{AB}$ being mixed and Alice’s and Bob’s measurement being POVMs. In such a general situation, the above measurement operators $A_{l|j}$ satisfy $A_{l|j}^{†}A_{l|j}\leq 𝟙$ and $A_{l|j}A_{l|j}^{†}\leq 𝟙$ for any *l* and *j*, and $A_{0|j}=𝟙$ for any *j* (the same holds for $B_{m|k}$ operators).

Since in our scenario we are dealing with three-outcome measurements, it is not difficult to observe that $A_{2|j}=A_{1|j}^{†}$. Therefore, by also taking into account the fact that $A_{0|j}=𝟙$, this implies that measurement is fully determined by a single operator $A_{1|j}$, which, for simplicity, we denote as $A_{j}$; analogously for Bob’s measurements, we denote $B_{k}\equiv B_{1|k}$. In the case of the measurements $F_{j}$ and $G_{k}$ being projective, $A_{l|j}$ and $B_{m|k}$ are all unitary operators whose spectra are $1,\omega ,\omega^{2}$, and can be represented as $A_{l|j}=A_{j}^{l}$ and $B_{m|k}=B_{k}^{m}$, where the superscripts *l* and *m* are operator powers of the unitary quantum observables $A_{j}$ and $B_{k}$, such that $A_{j}^{d}=B_{k}^{d}=𝟙$. In this case the expectation values (Equation 3) can be expressed as:(5)$$\langle A_{l|j}B_{m|k}\rangle =\langle \Psi_{ABE}|A_{j}^{l}⊗B_{k}^{m}⊗{𝟙}_{E}|\Psi_{ABE}\rangle .$$

Let us introduce now the Bell inequality used in our scenario. We consider a slightly simplified Bell operator from [17] for d=3, which is sufficient for our purposes. The modification results from the omission of the identity term, and now the Bell operator is given by:(6)$$W_{1}:=\frac{\lambda }{27}\sum_{j,k=0}^{2}\omega^{jk}A_{j}⊗B_{k}+h.c.,$$
where $\lambda =e^{-𝕚\pi /18}$, and h.c. stands for the hermitian conjugation. The corresponding Bell inequality is defined as:(7)$$\langle W_{1}\rangle \leq \beta_{L},$$
where $\beta_{L}$ is its classical bound, that is, the maximal value of the Bell expression $\langle W_{1}\rangle$ over all correlations admitting the local-realistic description, and it amounts to:$$\begin{array}{cc} \beta_{L} & =\frac{2\cos(\pi /9)}{3\sqrt{3}}. \end{array}$$

Moreover, the maximal quantum value of the above Bell inequality was found in Ref. [17] to be:(8)$$\beta_{Q}=\frac{2}{3\sqrt{3}}.$$

It is achieved by the maximally entangled state of two qutrits:(9)$$|\Phi_{AB}\rangle :=\frac{1}{\sqrt{3}}\sum_{j=0}^{2}{|i\rangle }_{A}{|i\rangle }_{B}$$
and the following choice of Bob’s observables:(10)$$B_{0}=Z,\;\;B_{1}=X,\;\;B_{2}=\omega X^{2}Z^{2},$$
where:(11)$$X:=\sum_{i=0}^{2}|i+1\rangle \;\;\langle i|\;and\;Z:=\sum_{i=0}^{2}\omega^{i}|i\rangle \;\;\langle i|$$
and |3〉≡|0〉. It is worth noticing that the eigenvectors of $B_{i}$ form mutually unbiased bases in $C^{3}$. At the same time, Alice’s optimal observables can be expressed as the following linear combinations of the above optimal observables of Bob:(12)$$A_{j}:=\frac{\lambda^{*}}{\sqrt{3}}\sum_{k}\omega^{-jk}B_{k}^{*},$$
where ${}^{*}$ denotes the complex conjugation in the standard basis.

The above Bell inequality can be used for device-independent characterization of Bob’s measurements. In fact, it follows from Ref. [17] that if it is maximally violated by correlations obtained from the first preparation $|\Psi_{ABE}^{1}\rangle$ and the measurements $B_{i}$(i=0,1,2), then the latter are projective. Moreover, $\dim\left(H_{B}\right)=3\cdot t_{B}$ for some positive integer $t_{B}$, or, equivalently, $H_{B}={\left(C^{3}\right)}_{B^{\prime }}⊗{\left(C^{t_{B}}\right)}_{B^{\prime \prime }}$, and there exists a unitary operation $U_{B}:H_{B}\to H_{B}$ such that:(13)$$\begin{array}{c} \begin{array}{cc} U_{B}B_{0}U_{B}^{†} & =Z⊗Q_{1}+Z⊗Q_{2},\\ U_{B}B_{1}U_{B}^{†} & =X⊗Q_{1}+X^{2}⊗Q_{2},\\ U_{B}B_{2}U_{B}^{†} & =\omega X^{2}Z^{2}⊗Q_{1}+XZ^{2}⊗Q_{2}, \end{array} \end{array}$$
where the operators $Q_{1},Q_{2}$ are orthogonal projectors satisfying $Q_{1}+Q_{2}={𝟙}_{B^{\prime \prime }}$. These two projectors identify the orthogonal subspaces corresponding to two inequivalent sets of observables maximally violating the Bell inequality (Equation 7) that are related via transposition.

Regarding the side of Alice, we can draw a similar conclusion as for Bob’s subsystem, and we can also determine the form of the first preparation $|\Psi_{ABE}^{1}\rangle$ [17]. However, we will not use their explicit forms in what follows; therefore, we do not present them here. In fact, the aim of performing the Bell test on the first preparation is to certify Bob’s measurements.

### 2.3. Steering Inequality

In this subsection, we recall the steering inequality introduced in Ref. [20], and we show how to use it for certification of the second preparation.

Let us again consider Alice and Bob performing measurements on some quantum state $|\Psi_{ABE}^{2}\rangle$. For a moment, assume that one of the measuring devices, let us say that belonging to Bob, is trusted and performs fixed measurements; Alice’s measurement device remains untrusted. Let us then consider the steering inequality constructed in Ref. [20]:(14)$$\langle W_{3}\rangle \leq {\tilde{\beta }}_{L},$$
where $W_{3}$ is a steering operator given by:(15)$$\begin{array}{ccc} W_{3} & = & A_{6}⊗B_{0}+\gamma A_{7}⊗B_{1}+\delta_{1}B_{0}+A_{6}^{†}⊗B_{0}^{†}+\gamma A_{7}^{†}⊗B_{1}^{†}+\delta_{1}^{*}B_{0}^{†}, \end{array}$$
where:(16)$$\gamma =3{\left(\sum_{\begin{array}{c} i,j=0\\ i\neq j \end{array}}^{2}\frac{\alpha_{i}}{\alpha_{j}}\right)}^{-1},\;\delta_{k}=-\frac{\gamma }{3}\sum_{\begin{array}{c} i,j=0\\ i\neq j \end{array}}^{2}\frac{\alpha_{i}}{\alpha_{j}}\omega^{-kj}.$$

Coefficients γ and $\delta_{k}$ are functions of three positive numbers $\alpha_{i}$ such that $\alpha_{1}^{2}+\alpha_{2}^{2}+\alpha_{3}^{2}=1$. Then, ${\tilde{\beta }}_{L}$ is the classical bound of (Equation 14). Although we do not know its explicit form for any $\alpha_{i}$, it was proven in Ref. [20] that for any $\alpha_{i}>0$ it is strictly lower than the maximal quantum value of $\langle W_{3}\rangle$, ${\tilde{\beta }}_{Q}=3$.

Recall also that $\alpha_{i}$ are Schmidt coefficients of the following state:(17)$$\begin{array}{c} |\psi (\alpha )\rangle =\sum_{i=0}^{2}\alpha_{i}|i_{A}\rangle |i_{B}\rangle , \end{array}$$
which maximally violates the inequality (Equation 7) for observables on the trusted side fixed to be *Z* and *X*, where we denoted $\alpha =(\alpha_{1},\alpha_{2},\alpha_{3})$.

Importantly, we can employ the above steering inequality to device-independently certify the second preparation $|\Psi_{ABE}^{2}\rangle$. To this end, we use it together with Bob’s observables $B_{0}$ and $B_{1}$, which, with the aid of the Bell inequality (Equation 6) (see Section 2.2), are certified to be of the form (Equation 13). Precisely, we can determine the form of Alice’s observables $A_{6}$ and $A_{7}$ in the sense that up to some unitary $U_{A}:H_{A}\to H_{A}$ we have:(18)$$U_{A}\;A_{6}\;U_{A}^{†}=Z⊗{𝟙}_{A^{\prime \prime }},\;U_{A}\;A_{7}\;U_{A}^{†}=X⊗{\bar{P}}_{1}+X^{2}⊗{\bar{P}}_{2},$$
such that ${\bar{P}}_{1},{\bar{P}}_{2}$ are orthogonal projectors satisfying ${\bar{P}}_{1}+{\bar{P}}_{2}={𝟙}_{A^{\prime \prime }}$. Moreover, the joint state corresponding to the second preparation $|\Psi_{ABE}^{2}\rangle$ up to the unitaries acting on Alice’s and Bob’s systems can be expressed as:(19)$$\left(U_{A}⊗U_{B}⊗{𝟙}_{E}\right)|\Psi_{ABE}^{2}\rangle ={|\psi (\alpha )\rangle }_{A^{\prime }B^{\prime }}⊗|\xi_{A^{\prime \prime }B^{\prime \prime }E}\rangle .$$

Therefore, the expression (Equation 19) constitutes a self-testing statement for any entangled state in dimension three.

## 3. Results

### 3.1. Certification of Full Weyl–Heisenberg Basis in d = 3

In this subsection we present a method for device-independent certification of a full W-H basis, which, up to some phases, consists of the operators $W_{p,q}:=X^{p}Z^{q}$, with p,q=0,1,2. We begin by observing that the eigenvectors of the particular subset of the W-H operators $W_{p,q}=X^{p}Z^{q}$, that is, $\left\{Z,X,XZ,XZ^{2}\right\}$ are mutually unbiased for d=3. Let $M_{r}$ denote subsequent elements of this subset, where r=0,1,2,3 and $\Pi_{r}^{\left(b\right)}$ denote the projectors constructed from the eigenvectors of $M_{r}$, where b=0,1,2 labels the outcomes of a given observable. Operators $M_{r}$ can be written in terms of the spectral decomposition as:(20)$$M_{r}=\sum_{b=0}^{2}\omega^{b}\Pi_{r}^{\left(b\right)}.$$

Let us note that multiplying the operators $M_{r}$ with powers of ω or taking conjugate transpose results just in relabeling $M_{r}$’s outcomes. Now let us observe that apart from the identity, the remaining W-H operators $\left\{Z^{2},X^{2},X^{2}Z,X^{2}Z^{2}\right\}$ can be obtained from the set $\left\{M_{r}\right\}$ through a suitable rearrangement of the eigenvalues:(21)$$\begin{array}{c} \begin{array}{cc} Z^{2}=M_{0}^{†} & =\sum_{b}\omega^{-b}\Pi_{0}^{\left(b\right)},\\ X^{2}=M_{1}^{†} & =\sum_{b}\omega^{-b}\Pi_{1}^{\left(b\right)},\\ X^{2}Z=\omega M_{3}^{†} & =\sum_{b}\omega^{-b+1}\Pi_{3}^{\left(b\right)},\\ X^{2}Z^{2}=\omega^{2}M_{2}^{†} & =\sum_{b}\omega^{-b+2}\Pi_{2}^{\left(b\right)}. \end{array} \end{array}$$

The conclusion from the above consideration is that it is enough to certify only 4 particular observables out of a full W-H basis, while the remaining ones can be obtained by adequately relabelling the outcomes.

As shown in Section 2.2, the maximal violation of the Bell inequality (Equation 7) allows one to certify three such particular observables. To certify the fourth one we consider another Bell operator given by:(22)$$\begin{array}{cc} W_{2}:=\frac{\lambda }{27}[A_{3}⊗\left(B_{0}+B_{2}^{†}+B_{3}\right)+A_{4}⊗(B_{0}+\omega B_{2}^{†} & +\omega^{2}B_{3})\\ & +A_{5}\left.⊗\left(B_{0}+\omega^{2}B_{2}^{†}+\omega B_{3}\right)\right]+h.c. \end{array}$$

Notice that while Alice’s measurements in this Bell operator are all different from those used in $W_{1}$, on Bob’s side the two measurements $B_{0}$ and $B_{2}$ are same.

We aim to prove that observation of the maximum violation of the Bell inequality (Equation 7) by the two different sets of observables corresponding to the Bell operators $W_{1}$ and $W_{2}$ self-tests the complete W-H basis in dimension three up to the unitary and transposition equivalences. First, maximal violation of (Equation 7) by $W_{1}$ implies that Bob’s measurements are projective and that the corresponding observables $B_{i}$ with i=0,1,2 are of the form (Equation 13). Second, it follows from Ref. [17] that maximal violation of the same inequality by $W_{2}$ implies that $B_{3}$ is a quantum observable too and that $B_{0}$, $B_{2}^{†}$ and $B_{3}$ must satisfy the following relations:(23)$$\begin{array}{c} \begin{array}{c} B_{0}^{†}=-\omega \left\{B_{2}^{†},B_{3}\right\},\\ B_{3}^{†}=-\omega \left\{B_{0},B_{2}^{†}\right\},\\ B_{2}=-\omega \left\{B_{3},B_{0}\right\}. \end{array} \end{array}$$

These conditions can be used to reconstruct the fourth observable of Bob, $B_{3}$. Precisely, plugging the forms of $B_{0}$ and $B_{2}$ into the second relation in (Equation 23), one immediately finds that:(24)$$U_{B}B_{3}U_{B}^{†}=\omega^{2}X^{2}Z⊗Q_{1}+XZ⊗Q_{2}.$$

Let us refer here to the fact that we cannot distinguish between optimal measurements used in the device-independent scenario and their transpositions based only on the correlations $\vec{p}$ observed in a Bell experiment. We call this ambiguity transposition inequivalence. For this reason, we are unable to perform a full tomography of the POVM on Alice’s side. In other words, it is not possible to completely reconstruct the elements of the POVM implemented on Alice’s side from the joint correlations. Consequently, we cannot certify the maximum randomness directly from the POVM elements. Nevertheless, in the following section, we present a way to overcome this difficulty. We show how to certify maximal randomness from the POVM on Alice’s side without certifying the measurement itself.

To summarize this section, we managed to certify four observables on Bob’s side by using the two Bell operators $W_{1}$ and $W_{2}$. Then, we observe that with a proper relabeling (Equation 21) we can construct the other four elements of the W-H basis. With the identity operator, we can then have the complete set of nine W-H operators in dimension three. For the convenience of further calculations, up to certain relabeling of the outcomes, let us express the certified observables $B_{k}$ as follows:(25)$$\begin{array}{c} U_{B}\;B_{p,q}\;U_{B}^{†}=X^{p}Z^{q}⊗Q_{1}+Z^{q}X^{2p}⊗Q_{2}, \end{array}$$
where p,q=0,1,2.

### 3.2. Certification of Randomness

We can finally proceed to the randomness certification. As discussed before, the last ingredient of our scheme is the nine-outcome POVM that Alice performs on her subsystem. Since the state on which $F_{8}$ acts is certified to be (Equation 19), without loss of generality, we can consider this measurement to be a POVM acting on a state of dimension 3·t where *t* is some positive integer. Let us denote the POVM under consideration by $\left\{{\tilde{R}}_{a}\right\}$, where a=0,1,⋯,8 represents the outcomes of the measurement. The correlations between the outcomes of non-ideal POVM and Bob’s observables $B_{p,q}$ (Equation 25) should equal those of the ideal setup, as expressed in Equation (Equation 1). This means that:(26)$$\begin{array}{c} \langle \Psi_{ABE}^{2}|{\tilde{R}}_{a}⊗B_{p,q}⊗{𝟙}_{E}|\Psi_{ABE}^{2}\rangle =\langle \psi (\alpha )|R_{a}⊗W_{p,q}|\psi (\alpha )\rangle , \end{array}$$
where $\left\{R_{a}\right\}$ represents some ideal extremal POVM. Let us recall that POVMs of a fixed number of outcomes form a convex set, and given that POVM is called extremal if it cannot be decomposed as a convex mixture of other POVMs, we will now demonstrate that the above construction provides the certification of maximal amount of local randomness from the non-ideal POVM.

Let us note here that the ideal POVM elements $R_{a}$ used in definition (Equation 26) can be written as:(27)$$R_{a}=\sum_{k,l=0}^{d-1}r_{k,l}^{a}P^{-1}{\left(X^{k}Z^{l}\right)}^{*}P^{-1},$$
which follows from the fact that $R_{a}$ belongs to a three-dimensional Hilbert space and $P=\sum_{i=0}^{2}\alpha_{i}|i\rangle \;\;\langle i|$, such that $\alpha_{i}\neq 0$ for all *i*. For a remark, let us note that operators $P^{-1}{\left(X^{k}Z^{l}\right)}^{*}P^{-1}$ form a complete operator basis for measurements acting on three-dimensional Hilbert space. Using the fact that $|\psi (\alpha )\rangle =\sqrt{3}P⊗{𝟙}_{B}|\Phi \rangle$, where:(28)$$|\Phi \rangle =\frac{1}{\sqrt{3}}\left(|00\rangle +|11\rangle +|22\rangle \right),$$
let us now look at the right-hand side of Equation (Equation 26), which, after a simple computation, gives:(29)$$\langle \psi (\alpha )|R_{a}⊗W_{p,q}|\psi (\alpha )\rangle =r_{p,q}^{a}.$$

Hence, from (Equation 26) we have that:(30)$$\langle \Psi_{ABE}|{\tilde{R}}_{a}⊗B_{p,q}⊗{𝟙}_{E}|\Psi_{ABE}\rangle =r_{p,q}^{a}.$$

Next we can substitute (Equation 19) into (Equation 30) and obtain:(31)$$\begin{array}{c} \begin{array}{c} \langle \xi_{A^{\prime \prime }B^{\prime \prime }E}|\langle \psi (\alpha )|U_{A}{\tilde{R}}_{a}U_{A}^{†}⊗U_{B}B_{p,q}U_{B}^{†}⊗{𝟙}_{E}|\psi (\alpha )\rangle |\xi_{A^{\prime \prime }B^{\prime \prime }E}\rangle =r_{p,q}^{a}, \end{array} \end{array}$$
where from now on, we will use the notation ${\bar{R}}_{a}=U_{A}{\tilde{R}}_{a}U_{A}^{†}$, and $U_{B}B_{p,q}U_{B}^{†}$ is given by Equation (Equation 25).

Since $\left\{{\bar{R}}_{a}\right\}$ acts on a subsystem of dimension 3·t, without loss of generality, we can decompose its elements as:(32)$${\bar{R}}_{a}=\sum_{k,l=0}^{2}P^{-1}{\left(X^{k}Z^{l}\right)}^{*}P^{-1}⊗{\bar{R}}_{k,l}^{a},$$
where ${\bar{R}}_{k,l}^{a}$ act on $H_{A^{\prime \prime }}$. Now we can insert an expression for ${\bar{R}}_{a}$ (Equation 32), the state $|\Psi_{ABE}^{2}\rangle$ from (Equation 19), and Bob’s measurements $B_{p,q}$ (Equation 25) into Equation (Equation 30) to find the following formulas:(33)$$\begin{array}{c} \langle \xi_{A^{\prime \prime }B^{\prime \prime }E}|{\bar{R}}_{0,q}^{a}⊗{𝟙}_{B^{\prime \prime }E}|\xi_{A^{\prime \prime }B^{\prime \prime }E}\rangle =r_{0,q}^{a} \end{array}$$
for p=0 and all *q*, and
(34)$$\begin{array}{cc} \langle \xi_{A^{\prime \prime }B^{\prime \prime }E}| & {\bar{R}}_{p,q}^{a}⊗Q_{1}⊗{𝟙}_{E}\\ \; & +\omega^{2pq}{\bar{R}}_{2p,q}^{a}⊗Q_{2}⊗{𝟙}_{E}|\xi_{A^{\prime \prime }B^{\prime \prime }E}\rangle =r_{p,q}^{a} \end{array}$$
for p=1,2 and all *q*. Inspired by Acín et al. [12], we define the following normalized states:(35)$$|\phi_{A^{\prime \prime }}^{b,e}\rangle =\frac{1}{\sqrt{q_{b,e}}}\left(𝟙⊗Q_{b}⊗\sqrt{Z_{e}}\right)|\xi_{A^{\prime \prime }B^{\prime \prime }E}\rangle ,$$
where b=1,2; $Z_{e}$ is a POVM element corresponding to Eve’s outcome *e* and $\sqrt{q_{b,e}}$ is a normalization factor. Now, by using $|\phi_{A^{\prime \prime }}^{b,e}\rangle$, we can reformulate coefficients (Equation 33) and (Equation 34) as follows:(36)$$r_{p,q}^{a}=\sum_{b=1,2}\sum_{e}q_{b,e}{\tilde{r}}_{p,q}^{a;b,e},$$
where we have defined the coefficients:(37)$${\tilde{r}}_{p,q}^{a;1,e}:=\langle \phi^{1,e}|{\bar{R}}_{p,q}^{a}⊗{𝟙}_{B^{\prime \prime }}⊗{𝟙}_{E}|\phi^{1,e}\rangle ,$$
and
(38)$${\tilde{r}}_{p,q}^{a;2,e}:=\omega^{2pq}\langle \phi^{2,e}|{\bar{R}}_{2p,q}^{a}⊗{𝟙}_{B^{\prime \prime }}⊗{𝟙}_{E}|\phi^{2,e}\rangle .$$

Let us now define the following operators in terms of coefficients (Equation 37) and (Equation 38):(39)$${\bar{R}}_{a}^{1,e}=\sum_{p,q}{\tilde{r}}_{p,q}^{a;1,e}P^{-1}{\left(X^{p}Z^{q}\right)}^{*}P^{-1},$$
and
(40)$${\bar{R}}_{a}^{2,e}=\sum_{p,q}{\tilde{r}}_{p,q}^{a;2,e}P^{-1}{\left(X^{p}Z^{q}\right)}^{*}P^{-1}.$$

It is easy to check that the operators ${\bar{R}}_{a}^{b,e}$ are valid POVMs, that is, they satisfy the following properties, ${\bar{R}}_{a}^{b,e}\geq 0$ and $\sum_{a}{\bar{R}}_{a}^{b,e}=𝟙$. To see this, using Equation (Equation 37) let us rewrite the decompositions of the operators given by Equation (Equation 39) as follows:(41)$${\bar{R}}_{a}^{1,e}={\mathrm{Tr}}_{A^{\prime \prime }B^{\prime \prime }E}\left[\left({\bar{R}}_{a}⊗{𝟙}_{B^{\prime \prime }E}\right)\left({𝟙}_{A}⊗|\phi^{1,e}\rangle \;\langle \phi^{1,e}|\right)\right].$$

From the fact that ${\bar{R}}_{a}\geq 0$ and $\sum_{a}{\bar{R}}_{a}=𝟙$ and from the above equation, it then follows that ${\bar{R}}_{a}^{1,e}\geq 0$ and $\sum_{a}{\bar{R}}_{a}^{1,e}=𝟙$. Therefore, the coefficients ${\tilde{r}}_{p,q}^{a;1,e}$ define a family of valid POVMs with the operators ${\bar{R}}^{1,e}$. Note that the operators ${\bar{R}}_{a}^{2,e}$ are a transposition of the operators ${\bar{R}}_{a}^{1,e}$. Thus, the coefficients ${\tilde{r}}_{p,q}^{a;2,e}$ also define a family of POVMs with the operators ${\bar{R}}^{2,e}$.

Using Equations (Equation 39) and (Equation 40), we have the following expression from Equation (Equation 36):(42)$$\begin{array}{c} R_{a}=\sum_{b,e}q_{b,e}{\bar{R}}_{a}^{b,e}, \end{array}$$
which can be understood as a convex decomposition of the ideal POVM $\left\{R_{a}\right\}$ in terms of the POVMs ${\bar{R}}^{b,e}$ with the respective weights $q_{b,e}$. However, the POVM $\left\{R_{a}\right\}$ is extremal and cannot be expressed as a convex combination of other POVMs. Thus, we have ${\tilde{r}}_{p,q}^{a;b,e}=r_{p,q}^{a}$ for all b,e and $\sum_{b,e}q_{b,e}=1$. Finally, rewriting the guessing probability (Equation 2) of Eve for outcomes of Alice’s POVM $\left\{{\bar{R}}_{a}\right\}$, we have:(43)$$\begin{array}{cc} G(j=8,\vec{p}) & =\sum_{a}\langle \Psi_{ABE}|{\bar{R}}_{a}⊗𝟙⊗Z_{a}|\Psi_{ABE}\rangle . \end{array}$$

Using Equations (Equation 19) and (Equation 32) in the above equation, we arrive at:(44)$$\begin{array}{cc} G(j=8,\vec{p}) & =\sum_{a}\langle \xi_{A^{\prime \prime }B^{\prime \prime }E}|{\bar{R}}_{0,0}^{a}⊗𝟙⊗Z_{a}|\xi_{A^{\prime \prime }B^{\prime \prime }E}\rangle . \end{array}$$

Now, using Equation (Equation 33), we can simplify (Equation 44) as follows:(45)$$\begin{array}{cc} G(j=8,\vec{p}) & =\sum_{b,a}q_{b,a}\;{\tilde{r}}_{0,0}^{a;b,a}=\sum_{b,a}q_{b,a}r_{0,0}^{a}, \end{array}$$
where we have used the constraint on ${\tilde{r}}_{0,0}^{a;b,a}$ as argued in the previous paragraph. Now, choosing an ideal POVM $\left\{R_{a}\right\}$ (Equation 27) such that
(46)$$\begin{array}{c} r_{0,0}^{a}=1/9\;\forall a, \end{array}$$
and using it for the guessing probability from Equation (Equation 45), gives $G(j=8,\vec{p})=1/9$. This implies that using the scheme mentioned above, Alice can securely generate $-\log_{2}G=2\log_{2}3$ bits of randomness from any partially entangled two-qutrit state provided that there exist extremal POVMs that satisfy the condition (Equation 46). As a final remark here let us note that Equation (Equation 29) for p,q=0,0 gives:(47)$$\langle \psi (\alpha )|R_{a}⊗{𝟙}_{B}|\psi (\alpha )\rangle =r_{0,0}^{a},$$
which is equivalent to the expression
(48)$$\mathrm{Tr}\left[R_{a}\rho_{A}\right]=r_{0,0}^{a},$$
where $\rho_{A}={\mathrm{Tr}}_{B}\left(|\psi (\alpha )\rangle \;\langle \psi (\alpha )|\right)$. In the following section we present the construction of the extremal POVM, which satisfies the condition (Equation 46).

### 3.3. Construction of Extremal Qutrit POVM

D’Ariano and collaborators [11] have classified all extremal POVMs with discrete output sets. According to this classification, an extremal POVM $\left\{R_{a}\right\}$ with $d^{2}$ outcomes must necessarily be rank-one, and its elements must be linearly independent. Linear independence of the POVM elements can be verified with the condition $\sum_{a}s_{a}R_{a}=O$, where O is the zero matrix, satisfied if and only if all $s_{a}=0$. We also require that POVM elements acting on Alice’s subsystem give equal probabilities, i.e., $\mathrm{Tr}[R_{a}\rho_{A}]=1/9$ for all *a* and $\rho_{A}$. Finding a general class of POVMs that meets all the above conditions for any partially entangled state proved to be a demanding task. Below we present a construction provided in Ref. [20] that fulfills the above requirements for a well-defined subset of entangled two-qutrit states.

The nine POVM elements are given by:(49)$$R_{a}:=\lambda_{a}|\alpha_{a}\rangle \;\;\langle \alpha_{a}|,$$
with
(50)$$\begin{array}{c} |\alpha_{0}\rangle =|0\rangle ,\;|\alpha_{8}\rangle =|2\rangle , \end{array}$$
and for a=1,⋯,7:$$\begin{array}{c} |\alpha_{a}\rangle =\mu_{0}|0\rangle +\mu_{1}\exp\left(\frac{2\pi 𝕚(a-1)}{7}\right)|1\rangle +\mu_{2}\exp\left(\frac{6\pi 𝕚(a-1)}{7}\right)|2\rangle . \end{array}$$
Moreover, the coefficients $\lambda_{a}$ are given by:(51)$$\begin{array}{cc} \lambda_{0} & =\frac{1}{9\alpha_{0}^{2}},\;\lambda_{2}=\frac{1}{9\alpha_{2}^{2}},\;\lambda_{a}=\frac{1}{7}\left(3-\lambda_{0}-\lambda_{2}\right), \end{array}$$
for a=1,⋯,7, whereas $\mu_{0},\mu_{1},\mu_{2}$ are are defined as:(52)$$\begin{array}{c} \mu_{0}=\sqrt{\frac{1-\lambda_{0}}{7\lambda_{1}}},\;\mu_{1}=\sqrt{\frac{1}{7\lambda_{1}}},\;\mu_{2}=\sqrt{\frac{1-\lambda_{2}}{7\lambda_{1}}}. \end{array}$$

Notice that the vectors $|\alpha_{a}\rangle$ are normalized.

To construct an extremal POVM, all of the above coefficients defining it must satisfy $0\;<\;\lambda_{i}\;<1\;$ and $0\;<\;\mu_{j}\;<\;1$, for i=0,⋯,8, and j=0,1,2. The above inequalities must be strict, as otherwise, if one of them equals 0 or 1, the operators $R_{b}$ are not linearly independent and thus do not form an extremal POVM. Solving the set of inequalities that emerge from the above constraints, one obtains $\alpha_{0}>1/3$ and $\alpha_{2}>1/3$. By relabeling the indices in (50) and (51) we can generalize the constraint on $\alpha_{0}$ and $\alpha_{2}$ to the requirement that any two out of three α’s have to be greater than 1/3. Let us recall at this point that up to local unitaries any bipartite state of local dimension three can be expressed using Schmidt decomposition in the form of Equation (Equation 17). Therefore, the above POVMs give rise to $2\log_{2}3$ bits of randomness if any two Schmidt coefficients of the state are greater than 1/3.

Using the Monte Carlo method [22], we have checked that the above condition is satisfied by 92.6% states out of $10^{7}$ randomly generated ones. It remains an open problem to find an extremal POVM that would generate maximal randomness from any entangled state in dimension three.

## 4. Discussion

In our work, we introduced a method for device-independent certification of maximal randomness from pure entangled states in dimension three using non-projective measurements. For this purpose, we exploited the extended Bell scenario introduced recently in Ref. [20], which combines device-independent certification of local measurements with one-sided device-independent certification of pure entangled states. In fact, we first showed how to certify the complete set of W-H operators in dimension three by using the self-testing scheme proposed in Ref. [17]. Then, by using the steering scenario proposed in [20], we provided certification of any entangled state in dimension three. These two components finally allowed us to achieve the main goal of the paper, that is, to certify $2\log_{2}3$ random bits by performing generalized measurements on Alice’s subsystem of partially-entangled two-qutrit states. In fact, we verified numerically that the constructed POVM that we used in our scheme covers a significant subset of 92.6% of bipartite entangled states.

Several interesting directions for further research emerge from our work. First, it would be extremely interesting to understand what is the maximal amount of global randomness that can be certified from entangled quantum states of local dimension *d* by performing non-projective measurements on both subsystems. Whereas the known theoretical limit says that at most $4\log_{2}d$ bits of randomness can be created in this way, it is unclear whether this limit is achievable. In fact, it was shown recently in Ref. [15] that in the case of two-qubit systems, 3.9527 bits of randomness is actually the maximal amount that can be generated in a device-independent way. This intriguing observation comes from the fact that an adversary can use some of the global correlations obtained with non-projective qubit measurements to infer information about the outcomes of a Bell experiment. Therefore, understanding the fundamental limit for generating global randomness from two-qudit states in the non-local scenario remains a very interesting challenge. Another interesting problem for further research is to provide constructions of $d^{2}$-outcome extremal POVMs that can be used to generate randomness from any, even arbitrarily small, entangled states.

## Figures and Tables

**Figure 1 entropy-24-00350-f001:**
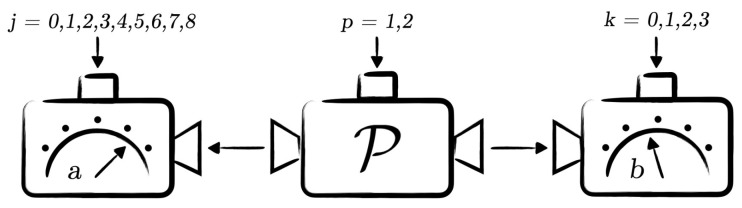
Randomness certification scenario for d=3. Alice and Bob have access to untrusted devices, and they apply measurements $F_{j}$ and $G_{k}$, respectively. The preparation box distributes two different bipartite states $\rho_{AB}^{1}$ for the preparation $P_{1}$ and $\rho_{AB}^{2}$ for the preparation $P_{2}$. After collecting the measurements statistics $\vec{p}$ one can device-independently certify Bob’s measurements based on Alice’s inputs j=0,1,2,3,4,5 and Bob’s inputs k=0,1,2,3. The preparation $P_{2}$ can be certified to be any pure entangled state of local dimension 3 with inputs j=6,7 and k=0,1. Randomness certification is based on the preparation $P_{2}$ and 9-outcome measurement corresponding to the input j=8.

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
