# Peer review of "Device-Independent Certification of Maximal Randomness from Pure Entangled Two-Qutrit States Using Non-Projective Measurements"

_entropy, 2022, doi:10.3390/e24030350_

Round 1

Reviewer 1 Report

This paper is well-written and presents a technically sound, precisely argued result. I have only some small corrections to suggest: 

  • I'm not sure it's necessary to abbreviate device-independent as DI - this decreases readability without any significant saving in space. It’s also not used consistently as sometimes the authors go back to writing `device-independent.' I recommend being consistent one way or another.
  • paragraph 2 page 2: `tranposition’ should be `transposition'
  • paragraph 2 page 2: it's reasonable enough to not waste space writing out what is meant by `the standard equivalences’ but I think it would be a good idea to at least include a reference to an article where a list of the standard equivalences can be found, otherwise there is room for misunderstanding
  • paragraph 3 page 3: ‘potential Eve’s strategies’ should this be something like `potential strategies for Eve’?
  • paragraph 1 page 5: 'and now Bell operator is given by’ should be 'and now the Bell operator is given by’ 
  • page 5 after equation 9: 'Bob’s obesrvables’ should be ‘Bob's observables’
  • page 5 after equation 12: ‘and that there exist a unitary operation’ should be `and that there exists a unitary operation’
  • page 6 paragraph 1: 'as well as we can determine’ should perhaps be `and we can also determine' or similar
  • page 6 after equation 16: ‘Whereas we do not know' I think whereas is the wrong word here - should perhaps be ‘although we do not know’
  • page 6: ‘Importantly, as shown in Ref. [19] from the maximal violation of the steering inequality (15) and the observables B0, B1 certified to be (13)’ this sentence is quite confusing, perhaps rewrite it to be clearer
  • page 7 paragraph 1: 'were r = 0, 1, 2, 3’ I think this should be ‘where'
  • page 7 paragraph 2: ‘suitable rearrangement of the eigenvalues that is’ maybe replace ‘that is’ with a colon

Author Response

We would like to thank the reviewer for reading our manuscript carefully and pointing out the elements that can be improved. We have considered all the reviewer's comments in the revised version of our manuscript.

Point 1

Spelling mistakes

Response 1

We corrected all spelling mistakes marked by the reviewer.

Point 2

I'm not sure it's necessary to abbreviate device-independent as DI - this decreases readability without any significant saving in space. It’s also not used consistently as sometimes the authors go back to writing `device-independent.' I recommend being consistent one way or another.

Response 2

We decided to use full name device-independent throughout the whole manuscript consistently.  

Point 3

Paragraph 2 page 2: it's reasonable enough to not waste space writing out what is meant by `the standard equivalences’ but I think it would be a good idea to at least include a reference to an article where a list of the standard equivalences can be found, otherwise there is room for misunderstanding

Response 3

We added a reference ([19] in the revised version) in which the standard equivalences in self-testing statements are well explained.

Point 4

page 6: ‘Importantly, as shown in Ref. [19] from the maximal violation of the steering inequality (15) and the observables B0, B1 certified to be (13)’ this sentence is quite confusing, perhaps rewrite it to be clearer

Response 4

We agree with the reviewer and have accordingly modified this paragraph to present the idea behind self-testing of the second preparation in a more transparent way.   

Reviewer 2 Report

In this work, the authors consider the problem of certifying the maximal amount of local randomness from pure entangled states in dimension 3, in a device-independent way. To do so, they consider Positive Operator Valued Measurements (POVM) and a modified Bell scenario. In particular, they consider a source capable of preparing two different entangled states, depending on the parameter p. One of the users, Alice can measure the state with eight 3-outcome measurements and one 9-outcome measurement, while the second party, Bob can measure with four 3-outcome measurements. Using the first preparation they can certify the full Weyl-Heisenberg basis in dimension 3. Then, with the second preparation and a steering inequality, they are able to certify any entangled state in dimension three. Finally, they show how to build a POVM to certify the maximal amount of randomness for a subset of partially entangled states.

After reading the manuscript I have the following major comments:

  • In the introduction, the authors write “However, it remains an open and highly nontrivial problem whether it is possible to device-independently certify the maximal amount of 2 log2 d bits of randomness by performing measurements on quantum systems of dimension d for any finite d” and then related to this “We provide a positive answer to the first problem”. However, in the manuscript, the authors always consider the case of dimension 3. Can the author clarify this point?
  • The explicit construction of the POVM that satisfies condition 46 has been shown numerically not to hold in general for any partially-entangled states. I think it would be interesting to show for which classes of states this condition holds and for which it doesn’t. Moreover, I think it should be better highlighted in the introduction that this condition does not hold in general.

Based on the previous comments I cannot recommend the publication of the manuscript in Entropy in its current form.

Author Response

We thank the reviewer for reading and commenting on our manuscript. The reviewer noticed two points that should be improved. We have addressed these points in the revised version of the manuscript.

Point 1

In the introduction, the authors write “However, it remains an open and highly nontrivial problem whether it is possible to device-independently certify the maximal amount of 2 log2 d bits of randomness by performing measurements on quantum systems of dimension d for any finite d” and then related to this “We provide a positive answer to the first problem”. However, in the manuscript, the authors always consider the case of dimension 3. Can the author clarify this point?

Response 1

We have corrected this paragraph since, as the reviewer had pointed out, it could create confusion. We stated more clearly that we are solving the problem of maximal randomness certification only in dimension three and for a particular subset of entangled states. Solving this problem for any dimension remains a demanding open problem. 

Point 2

The explicit construction of the POVM that satisfies condition 46 has been shown numerically not to hold in general for any partially-entangled states. I think it would be interesting to show for which classes of states this condition holds and for which it doesn’t. Moreover, I think it should be better highlighted in the introduction that this condition does not hold in general.

Response 2

We clarify the construction of the extremal POVM with more details.
First, we explain the reasons for the limitations contained in our construction. Then, we explain how to analytically compute the constraints on Schmidt coefficients of the bipartite state shared between Alice and Bob. This constraint requires that two out of three Schmidt coefficients need to be greater than 1/3. By using a Monte Carlo method, we confirmed that the above constraint allows us to construct an extremal POVM generating maximal randomness for 92,6% of entangled two-qutrit states. 

Reviewer 3 Report

In this paper it is introduced a method for device-independent certification of the maximal possible amount of 2 log2 3 random bits using pure bipartite entangled two-qutrit states and extremal nine-outcome general non-projective measurements. The authors employ the extended Bell scenario, which combines a device-independent method for certification of the full Weyl-Heisenberg basis in three-dimensional Hilbert spaces and a one-sided device-independent method for certification of two-qutrit partially entangled states.

The paper is well and clearly written, and the obtained results could present interest in designing protocols for quantum information tasks, like quantum cryptography and quantum key distribution.

I conclude by appreciating that this manuscript can be published, in the present form, in the journal Entropy.

Author Response

Dear reviewer
We are grateful for reading our manuscript and for the positive evaluation of it.

Kind regards,

the authors

Round 2

Reviewer 2 Report

The authors have successfully replied to my comments, therefore I can recommend the publication of the manuscript in Entropy.